# A Scalable Approach to Probabilistic Latent Space Inference of Large-Scale Networks

**Junming Yin**
School of Computer Science
Carnegie Mellon University
Pittsburgh, PA 15213
junmingy@cs.cmu.edu

**Qirong Ho**
School of Computer Science
Carnegie Mellon University
Pittsburgh, PA 15213
qho@cs.cmu.edu

**Eric P. Xing**
School of Computer Science
Carnegie Mellon University
Pittsburgh, PA 15213
epxing@cs.cmu.edu

## Abstract

We propose a scalable approach for making inference about latent spaces of large networks. With a succinct representation of networks as a bag of triangular motifs, a parsimonious statistical model, and an efficient stochastic variational inference algorithm, we are able to analyze real networks with over a million vertices and hundreds of latent roles on a single machine in a matter of hours, a setting that is out of reach for many existing methods. When compared to the state-of-the-art probabilistic approaches, our method is several orders of magnitude faster, with competitive or improved accuracy for latent space recovery and link prediction.

## 1 Introduction

In the context of network analysis, a latent space refers to a space of unobserved latent representations of individual entities (i.e., topics, roles, or simply embeddings, depending on how users would interpret them) that govern the potential patterns of network relations. The problem of latent space inference amounts to learning the bases of such a space and reducing the high-dimensional network data to such a lower-dimensional space, in which each entity has a position vector. Depending on model semantics, the position vectors can be used for diverse tasks such as community detection [1, 5], user personalization [4, 13], link prediction [14] and exploratory analysis [9, 19, 8]. However, scalability is a key challenge for many existing probabilistic methods, as even recent state-of-the-art methods [5, 8] still require days to process modest networks of around $100,000$ nodes.

To perform latent space analysis on at least million-node (if not larger) real social networks with many distinct latent roles [24], one must design inferential mechanisms that scale in both the number of vertices $N$ and the number of latent roles $K$. In this paper, we argue that the following three principles are crucial for successful large-scale inference: (1) succinct but informative representation of networks; (2) parsimonious statistical modeling; (3) scalable and parallel inference algorithms. Existing approaches [1, 5, 7, 8, 14] are limited in that they consider only one or two of the above principles, and therefore can not simultaneously achieve scalability and sufficient accuracy. For example, the mixed-membership stochastic blockmodel (MMSB) [1] is a probabilistic latent space model for edge representation of networks. Its batch variational inference algorithm has $O(N^2 K^2)$ time complexity and hence cannot be scaled to large networks. The a-MMSB [5] improves upon MMSB by applying principles (2) and (3): it reduces the dimension of the parameter space from $O(K^2)$ to $O(K)$, and applies a stochastic variational algorithm for fast inference. Fundamentally, however, the a-MMSB still depends on the $\mathrm{O}(N^2)$ adjacency matrix representation of networks, just like the MMSB. The a-MMSB inference algorithm mitigates this issue by downsampling zero elements in the matrix, but is still not fast enough to handle networks with $N \geq 100,000$.

But looking beyond the edge-based relations and features, other higher-order structural statistics (such as the counts of triangles and k-stars) are also widely used to represent the probability distribution over the space of networks, and are viewed as crucial elements in building a good-fitting exponential random graph model (ERGM) [11]. These higher-order relations have motivated the development of the triangular representation of networks [8], in which each network is represented succinctly as a bag of triangular motifs with size typically much smaller than $\Theta(N^2)$. This succinct representation has proven effective in extracting informative mixed-membership roles from

networks with high fidelity, thus achieving the first principle (1). However, the corresponding statistical model, called the mixed-membership triangular model (MMTM), only scales well against the size of a network, but does not scale to large numbers of latent roles (i.e., dimension of the latent space). To be precise, if there are $K$ distinct latent roles, its tensor of triangle-generating parameters is of size $O(K^3)$, and its blocked Gibbs sampler requires $O(K^3)$ time per iteration. Our own experiments show that the MMTM Gibbs algorithm is unusable for $K > 10$.

We now present a scalable approach to both latent space modeling and inference algorithm design that encompasses all three aforementioned principles for large networks. Specifically, we build our approach on the bag-of-triangles representation of networks [8] and apply principles (2) and (3), yielding a fast inference procedure that has time complexity $O(NK)$. In Section 3, we propose the parsimonious triangular model (PTM), in which the dimension of the triangle-generating parameters only grows linearly in $K$. The dramatic reduction is principally achieved by sharing parameters among certain groups of latent roles. Then, in Section 4, we develop a fast stochastic natural gradient ascent algorithm for performing variational inference, where an unbiased estimate of the natural gradient is obtained by subsampling a "mini-batch" of triangular motifs. Instead of adopting a fully factorized, naive mean-field approximation, which we find performs poorly in practice, we pursue a structured mean-field approach that captures higher-order dependencies between latent variables. These new developments all combine to yield an efficient inference algorithm that usually converges after 2 passes on each triangular motif (or up to 4-5 passes at worst), and achieves competitive or improved accuracy for latent space recovery and link prediction on synthetic and real networks. Finally, in Section 5, we demonstrate that our algorithm converges and infers a 100-role latent space on a 1M-node Youtube social network in just 4 hours, using a single machine with 8 threads.

## 2 Triangular Representation of Networks

We take a scalable approach to network modeling by representing each network succinctly as a bag of triangular motifs [8]. Each triangular motif is a *connected* subgraph over a vertex triple containing 2 or 3 edges (called open triangle and closed triangle respectively). Empty and single-edge triples are ignored. Although this triangular format does not preserve all network information found in an edge representation, these three-node connected subgraphs are able to capture a number of informative structural features in the network. For example, in social network theory, the notion of triadic closure [21, 6] is commonly measured by the relative number of closed triangles compared to the total number of connected triples, known as the global clustering coefficient or transitivity [17]. The same quantity is treated as a general network statistic in the exponential random graph model (ERGM) literature [16]. Furthermore, the most significant and recurrent structural patterns in many complex networks, so-called "network motifs", turn out to be connected three-node subgraphs [15].

Most importantly of all, triangular modeling requires much less computational cost compared to edge-based models, with little or no degradation of performance for latent space recovery [8]. In networks with $N$ vertices and low maximum vertex degree $\mathbf{D}$, the number of triangular motifs $\Theta(N\mathbf{D}^2)$ is normally much smaller than $\Theta(N^2)$, allowing us to construct more efficient inference algorithms scalable to larger networks. For high-maximum-degree networks, the triangular motifs can be subsampled in a node-centric fashion as a local data reduction step. For each vertex $i$ with degree higher than a user-chosen threshold $\delta$, uniformly sample $\binom{\delta}{2}$ triangles from the set composed of (a) its adjacent closed triangles, and (b) its adjacent open triangles that are *centered* on $i$. Vertices with degree $\leq \delta$ keep all triangles from their set. It has been shown that this $\delta$-subsampling procedure can approximately preserve the distribution over open and closed triangles, and allows for much faster inference algorithms (linear growth in N) at a small cost in accuracy [8].

In what follows, we assume that a preprocessing step has been performed — namely, extracting and $\delta$-subsampling triangular motifs (which can be done in $O(1)$ time per sample, and requires $< 1\%$ of the actual inference time) — to yield a bag-of-triangles representation of the input network. For each triplet of vertices $i, j, k \in \{1, \ldots, N\}, i < j < k$, let $E_{ijk}$ denote the observed type of triangular motif formed among these three vertices: $E_{ijk} = 1, 2$ and $3$ represent an open triangle with $i$, $j$ and $k$ in the center respectively, and $E_{ijk} = 4$ if a closed triangle is formed. Because empty and single-edge triples are discarded, the set of triples with triangular motifs formed, $I = \{(i, j, k) : i < j < k, E_{ijk} = 1, 2, 3 \text{ or } 4\}$, is of size $O(N\delta^2)$ after $\delta$-subsampling [8].

## 3 Parsimonious Triangular Model

Given the input network, now represented as a bag of triangular motifs, our goal is to make inference about the latent position vector $\theta_i$ of each vertex $i \in \{1, \ldots, N\}$. We take a mixed-membership

| $(s_{i,jk}, s_{j,ik}, s_{k,ij})$ | Equivalence classes | Conditional probability of $E_{ijk} \in \{1, 2, 3, 4\}$ |
|---|---|---|
| $x = s_{i,jk} = s_{j,ik} = s_{k,ij}$ | $\{1,2,3\}, \{4\}$ | $\text{Discrete}\left(\left[\frac{B_{xxx,1}}{3}, \frac{B_{xxx,1}}{3}, \frac{B_{xxx,1}}{3}, B_{xxx,2}\right]\right)$ |
| $x = s_{i,jk} = s_{j,ik} \neq s_{k,ij}$ | $\{1,2\}, \{3\}, \{4\}$ | $\text{Discrete}\left(\left[\frac{B_{xx,1}}{2}, \frac{B_{xx,1}}{2}, B_{xx,2}, B_{xx,3}\right]\right)$ |
| $x = s_{i,jk} = s_{k,ij} \neq s_{j,ik}$ | $\{1,3\}, \{2\}, \{4\}$ | $\text{Discrete}\left(\left[\frac{B_{xx,1}}{2}, B_{xx,2}, \frac{B_{xx,1}}{2}, B_{xx,3}\right]\right)$ |
| $x = s_{j,ik} = s_{k,ij} \neq s_{i,jk}$ | $\{2,3\}, \{1\}, \{4\}$ | $\text{Discrete}\left(\left[B_{xx,2}, \frac{B_{xx,1}}{2}, \frac{B_{xx,1}}{2}, B_{xx,3}\right]\right)$ |
| $s_{k,ij} \neq s_{i,jk} \neq s_{j,ik}$ | $\{1,2,3\}, \{4\}$ | $\text{Discrete}\left(\left[\frac{B_{0,1}}{3}, \frac{B_{0,1}}{3}, \frac{B_{0,1}}{3}, B_{0,2}\right]\right)$ |

Table 1: Equivalence classes and conditional probabilities of $E_{ijk}$ given $s_{i,jk}, s_{j,ik}, s_{k,ij}$ (see text for details).

approach: each vertex $i$ can take a mixture distribution over $K$ latent roles governed by a mixed-membership vector $\theta_i \in \Delta^{K-1}$ restricted to the $(K-1)$-simplex. Such vectors can be used for performing community detection and link prediction, as demonstrated in Section 5. Following a design principle similar to the Mixed-Membership Triangular Model (MMTM) [8], our Parsimonious Triangular Model (PTM) is essentially a latent-space model that defines the generative process for a bag of triangular motifs. However, compared to the MMTM, the major advantage of the PTM lies in its more compact and lower-dimensional nature that allows for more efficient inference algorithms (see Global Update step in Section 4). The dimension of triangle-generating parameters in the PTM is just $O(K)$, rather than $O(K^3)$ in the MMTM (see below for further discussion).

To form a triangular motif $E_{ijk}$ for each triplet of vertices $(i, j, k)$, a triplet of role indices $s_{i,jk}, s_{j,ik}, s_{k,ij} \in \{1, \ldots, K\}$ is first chosen based on the mixed-membership vectors $\theta_i, \theta_j, \theta_k$. These indices designate the roles taken by each vertex participating in this triangular motif. There are $O(K^3)$ distinct configurations of such latent role triplet, and the MMTM uses a tensor of triangle-generating parameters of the same size to define the probability of $E_{ijk}$, one entry $B_{xyz}$ for each possible configuration $(x, y, z)$. In the PTM, we reduce the number of such parameters by partitioning the $O(K^3)$ configuration space into several groups, and then sharing parameters within the same group. The partitioning is based on the number of distinct states in the configuration of the role triplet: 1) if the three role indices are all in the same state $x$, the triangle-generating probability is determined by $B_{xxx}$; 2) if only two role indices exhibit the same state $x$ (called majority role), the probability of triangles is governed by $B_{xx}$, which is shared across different minority roles; 3) if the three role indices are all distinct, the probability of triangular motifs depends on $B_0$, a single parameter independent of the role configurations. This sharing yields just $O(K)$ parameters $B_0, B_{xx}, B_{xxx}, x \in \{1, \ldots, K\}$, allowing PTM to scale to far more latent roles than MMTM. A similar idea was proposed in a-MMSB [5], using one parameter $\epsilon$ to determine inter-role link probabilities, rather than $O(K^2)$ parameters for all pairs of distinct roles, as in the original MMSB [1].

Once the role triplet $(s_{i,jk}, s_{j,ik}, s_{k,ij})$ is chosen, some of the triangular motifs can become indistinguishable. To illustrate, in the case of $x = s_{i,jk} = s_{j,ik} \neq s_{k,ij}$, one cannot distinguish the open triangle with $i$ in the center ($E_{ijk} = 1$) from that with $j$ in the center ($E_{ijk} = 2$), because both are open triangles centered at a vertex with majority role $x$, and are thus *structurally equivalent* under the given role configuration. Formally, this configuration induces a set of triangle equivalence classes $\{\{1, 2\}, \{3\}, \{4\}\}$ of all possible triangular motifs $\{1, 2, 3, 4\}$. We treat the triangular motifs within the same equivalence class as *stochastically equivalent*; that is, the conditional probabilities of events $E_{ijk} = 1$ and $E_{ijk} = 2$ are the same if $x = s_{i,jk} = s_{j,ik} \neq s_{k,ij}$. All possible cases are enumerated as follows (see also Table 1):

**1.** If all three vertices have the same role $x$, all three open triangles are equivalent and the induced set of equivalence classes is $\{\{1, 2, 3\}, \{4\}\}$. The probability of $E_{ijk}$ is determined by $B_{xxx} \in \Delta^1$, where $B_{xxx,1}$ represents the *total* probability of sampling an open triangle from $\{1, 2, 3\}$ and $B_{xxx,2}$ represents the closed triangle probability. Thus, the probability of a particular open triangle is $B_{xxx,1}/3$.

**2.** If only two vertices have the same role $x$ (majority role), the probability of $E_{ijk}$ is governed by $B_{xx} \in \Delta^2$. Here, $B_{xx,1}$ and $B_{xx,2}$ represent the open triangle probabilities (for open triangles centered at a vertex in majority and minority role respectively), and $B_{xx,3}$ represents the closed triangle probability. There are two possible open triangles with a vertex in majority role at the center, and hence each has probability $B_{xx,1}/2$.

**3.** If all three vertices have distinct roles, the probability of $E_{ijk}$ depends on $B_0 \in \Delta^1$, where $B_{0,1}$ represents the *total* probability of sampling an open triangle from $\{1, 2, 3\}$ (regardless of the center vertex's role) and $B_{0,2}$ represents the closed triangle probability.

To summarize, the PTM assumes the following generative process for a bag of triangular motifs:

- Choose $B_0 \in \Delta^1$, $B_{xx} \in \Delta^2$ and $B_{xxx} \in \Delta^1$ for each role $x \in \{1, \ldots, K\}$ according to symmetric Dirichlet distributions $\text{Dirichlet}(\lambda)$.

- For each vertex $i \in \{1, \ldots, N\}$, draw a mixed-membership vector $\theta_i \sim \text{Dirichlet}(\alpha)$.
- For each triplet of vertices $(i, j, k)$, $i < j < k$,
  - Draw role indices $s_{i,jk} \sim \text{Discrete}(\theta_i)$, $s_{j,ik} \sim \text{Discrete}(\theta_j)$, $s_{k,ij} \sim \text{Discrete}(\theta_k)$.
  - Choose a triangular motif $E_{ijk} \in \{1, 2, 3, 4\}$ based on $B_0, B_{xx}, B_{xxx}$ and the configuration of $(s_{i,jk}, s_{j,ik}, s_{k,ij})$ (see Table 1 for the conditional probabilities).

It is worth pointing out that, similar to the MMTM, our PTM is not a generative model of networks *per se* because (a) empty and single-edge motifs are not modeled, and (b) one can generate a set of triangles that does not correspond to any network, because the generative process does not force overlapping triangles to have consistent edge values. However, given a bag of triangular motifs $\mathbf{E}$ extracted from a network, the above procedure defines a valid probabilistic model $p(\mathbf{E} \mid \alpha, \lambda)$ and we can legitimately use it for performing posterior inference $p(\mathbf{s}, \theta, \mathbf{B} \mid \mathbf{E}, \alpha, \lambda)$. We stress that our goal is latent space inference, not network simulation.

## 4  Scalable Stochastic Variational Inference

In this section, we present a stochastic variational inference algorithm [10] for performing approximate inference under our model. Although it is also feasible to develop such algorithm for the MMTM [8], the $O(NK^3)$ computational complexity precludes its application to large numbers of latent roles. However, due to the parsimonious $O(K)$ parameterization of the PTM, our efficient algorithm has only $O(NK)$ complexity.

We adopted a structured mean-field approximation method, in which the true posterior of latent variables $p(\mathbf{s}, \theta, \mathbf{B} \mid \mathbf{E}, \alpha, \lambda)$ is approximated by a *partially* factorized distribution $q(\mathbf{s}, \theta, \mathbf{B})$,

$$q(\mathbf{s}, \theta, \mathbf{B}) = \prod_{(i,j,k) \in I} q(s_{i,jk}, s_{j,ik}, s_{k,ij} \mid \phi_{ijk}) \prod_{i=1}^{N} q(\theta_i \mid \gamma_i) \prod_{x=1}^{K} q(B_{xxx} \mid \eta_{xxx}) \prod_{x=1}^{K} q(B_{xx} \mid \eta_{xx}) q(B_0 \mid \eta_0),$$

where $I = \{(i, j, k) : i < j < k, E_{ijk} = 1, 2, 3 \text{ or } 4\}$ and $|I| = O(N\delta^2)$. The strong dependencies among the per-triangle latent roles $(s_{i,jk}, s_{j,ik}, s_{k,ij})$ suggest that we should model them as a group, rather than completely independent as in a naive mean-field approximation[1]. Thus, the variational posterior of $(s_{i,jk}, s_{j,ik}, s_{k,ij})$ is the discrete distribution

$$q(s_{i,jk} = x, s_{j,ik} = y, s_{k,ij} = z) \doteq q_{ijk}(x, y, z) = \phi_{ijk}^{xyz}, \quad x, y, z = 1, \ldots, K. \tag{1}$$

The posterior $q(\theta_i)$ is a Dirichlet($\gamma_i$); and the posteriors of $B_{xxx}, B_{xx}, B_0$ are parameterized as: $q(B_{xxx}) = \text{Dirichlet}(\eta_{xxx}), q(B_{xx}) = \text{Dirichlet}(\eta_{xx})$, and $q(B_0) = \text{Dirichlet}(\eta_0)$.

The mean field approximation aims to minimize the KL divergence $\text{KL}(q \parallel p)$ between the approximating distribution $q$ and the true posterior $p$; it is equivalent to maximizing a lower bound $\mathcal{L}(\phi, \eta, \gamma)$ of the log marginal likelihood of the triangular motifs (based on Jensen's inequality) with respect to the variational parameters $\{\phi, \eta, \gamma\}$ [22].

$$\log p(\mathbf{E} \mid \alpha, \lambda) \geq \mathbb{E}_q[\log p(\mathbf{E}, \mathbf{s}, \theta, \mathbf{B} \mid \alpha, \lambda)] - \mathbb{E}_q[\log q(\mathbf{s}, \theta, \mathbf{B})] \doteq \mathcal{L}(\phi, \eta, \gamma). \tag{2}$$

To simplify the notation, we decompose the variational objective $\mathcal{L}(\phi, \eta, \gamma)$ into a global term and a summation of local terms, one term for each triangular motif (see Appendix for details).

$$\mathcal{L}(\phi, \eta, \gamma) = g(\eta, \gamma) + \sum_{(i,j,k) \in I} \ell(\phi_{ijk}, \eta, \gamma). \tag{3}$$

The global term $g(\eta, \gamma)$ depends only on the global variational parameters $\eta$, which govern the posterior of the triangle-generating probabilities $\mathbf{B}$, as well as the per-node mixed-membership parameters $\gamma$. Each local term $\ell(\phi_{ijk}, \eta, \gamma)$ depends on per-triangle parameters $\phi_{ijk}$ as well as the global parameters. Define $\mathcal{L}(\eta, \gamma) \doteq \max_\phi \mathcal{L}(\phi, \eta, \gamma)$, which is the variational objective achieved by fixing the global parameters $\eta, \gamma$ and optimizing the local parameters $\phi$. By equation (3),

$$\mathcal{L}(\eta, \gamma) = g(\eta, \gamma) + \sum_{(i,j,k) \in I} \max_{\phi_{ijk}} \ell(\phi_{ijk}, \eta, \gamma). \tag{4}$$

Stochastic variational inference is a stochastic gradient ascent algorithm [3] that maximizes $\mathcal{L}(\eta, \gamma)$, based on noisy estimates of its gradient with respect to $\eta$ and $\gamma$. Whereas computing the true gradient $\nabla \mathcal{L}(\eta, \gamma)$ involves a costly summation over all triangular motifs as in (4), an unbiased noisy approximation of the gradient can be obtained much more cheaply by summing over a small subsample of triangles. With this unbiased estimate of the gradient and a suitable adaptive step size, the algorithm is guaranteed to converge to a stationary point of the variational objective $\mathcal{L}(\eta, \gamma)$ [18].

**Algorithm 1** Stochastic Variational Inference

1: $t = 0$. Initialize the global parameters $\boldsymbol{\eta}$ and $\boldsymbol{\gamma}$.
2: Repeat the following steps until convergence.
    (1) Sample a mini-batch of triangles $S$.
    (2) Optimize the local parameters $q_{ijk}(x, y, z)$ for all sampled triangles in parallel by (6).
    (3) Accumulate sufficient statistics for the natural gradients of $\boldsymbol{\eta}, \boldsymbol{\gamma}$ (and then discard $q_{ijk}(x, y, z)$).
    (4) Optimize the global parameters $\boldsymbol{\eta}$ and $\boldsymbol{\gamma}$ by the stochastic natural gradient ascent rule (7).
    (5) $\rho_t \leftarrow \tau_0(\tau_1 + t)^{-\kappa}$, $t \leftarrow t + 1$.

In our setting, the most natural way to obtain an unbiased gradient of $\mathcal{L}(\boldsymbol{\eta}, \boldsymbol{\gamma})$ is to sample a "mini-batch" of triangular motifs at each iteration, and then average the gradient of local terms in (4) *only* for these sampled triangles. Formally, let $m$ be the total number of triangles and define

$$\mathcal{L}_S(\boldsymbol{\eta}, \boldsymbol{\gamma}) = g(\boldsymbol{\eta}, \boldsymbol{\gamma}) + \frac{m}{|S|} \sum_{(i,j,k) \in S} \max_{\phi_{ijk}} \ell(\phi_{ijk}, \boldsymbol{\eta}, \boldsymbol{\gamma}), \tag{5}$$

where $S$ is a mini-batch of triangles sampled uniformly at random. It is easy to verify that $\mathbb{E}_S[\mathcal{L}_S(\boldsymbol{\eta}, \boldsymbol{\gamma})] = \mathcal{L}(\boldsymbol{\eta}, \boldsymbol{\gamma})$, hence $\nabla \mathcal{L}_S(\boldsymbol{\eta}, \boldsymbol{\gamma})$ is unbiased: $\mathbb{E}_S[\nabla \mathcal{L}_S(\boldsymbol{\eta}, \boldsymbol{\gamma})] = \nabla \mathcal{L}(\boldsymbol{\eta}, \boldsymbol{\gamma})$.

**Exact Local Update.** To obtain the gradient $\nabla \mathcal{L}_S(\boldsymbol{\eta}, \boldsymbol{\gamma})$, one needs to compute the optimal local variational parameters $\phi_{ijk}$ (keeping $\boldsymbol{\eta}$ and $\boldsymbol{\gamma}$ fixed) for each sampled triangle $(i, j, k)$ in the mini-batch $S$; these optimal $\phi_{ijk}$'s are then used in equation (5) to compute $\nabla \mathcal{L}_S(\boldsymbol{\eta}, \boldsymbol{\gamma})$. Taking partial derivatives of (3) with respect to each local term $\phi_{ijk}^{xyz}$ and setting them to zero, we get for distinct $x, y, z \in \{1, \ldots, K\}$,
$$\phi_{ijk}^{xyz} \propto \exp \left\{ \mathbb{E}_q[\log B_{0,2}] \mathbb{I}[E_{ijk} = 4] + \mathbb{E}_q[\log(B_{0,1}/3)] \mathbb{I}[E_{ijk} \neq 4] + \mathbb{E}_q[\log \theta_{i,x} + \log \theta_{j,x} + \log \theta_{k,x}] \right\}.$$
$$\tag{6}$$
See Appendix for the update equations of $\phi_{ijk}^{xxx}$ and $\phi_{ijk}^{xxy}$ ($x \neq y$).

$\mathrm{O}(K)$ **Approximation to Local Update.** For each sampled triangle $(i, j, k)$, the exact local update requires $\mathrm{O}(K^3)$ work to solve for all $\phi_{ijk}^{xyz}$, making it unscalable. To enable a faster local update, we replace $q_{ijk}(x, y, z \mid \phi_{ijk})$ in (1) with a simpler "mixture-of-deltas" variational distribution,

$$q_{ijk}(x, y, z \mid \delta_{ijk}) = \sum_a \delta_{ijk}^{aaa} \mathbb{I}[x = y = z = a] + \sum_{(a,b,c) \in \mathcal{A}} \delta_{ijk}^{abc} \mathbb{I}[x = a, y = b, z = c],$$

where $\mathcal{A}$ is a randomly chosen set of triples $(a, b, c)$ with size $\mathrm{O}(K)$, and $\sum_a \delta_{ijk}^{aaa} + \sum_{(a,b,c) \in \mathcal{A}} \delta_{ijk}^{abc} = 1$. In other words, we assume the probability mass of the variational posterior $q(s_{i,jk}, s_{j,ik}, s_{k,ij})$ falls entirely on the $K$ "diagonal" role combinations $(a, a, a)$ as well as $\mathrm{O}(K)$ randomly chosen "off-diagonals" $(a, b, c)$. Conveniently, the $\delta$ update equations are identical to their $\phi$ counterparts as in (6), except that we normalize over the $\delta$'s instead.

In our implementation, we generate $\mathcal{A}$ by picking $3K$ combinations of the form $(a, a, b)$, $(a, b, a)$ or $(a, a, b)$, and another $3K$ combinations of the form $(a, b, c)$, thus mirroring the parameter structure of $\mathbf{B}$. Furthermore, we re-pick $\mathcal{A}$ every time we perform the local update on some triangle $(i, j, k)$, thus avoiding any bias due to a single choice of $\mathcal{A}$. We find that this approximation works as well as the full parameterization in (1), yet requires only $\mathrm{O}(K)$ work per sampled triangle. Note that any choice of $\mathcal{A}$ yields a valid lower bound to the true log-likelihood; this follows from standard variational inference theory.

**Global Update.** We appeal to stochastic natural gradient ascent [2, 20, 10] to optimize the global parameters $\boldsymbol{\eta}$ and $\boldsymbol{\gamma}$, as it greatly simplifies the update rules while maintaining the same asymptotic convergence properties as classical stochastic gradient. The natural gradient $\tilde{\nabla} \mathcal{L}_S(\boldsymbol{\eta}, \boldsymbol{\gamma})$ is obtained by a premultiplication of the ordinary gradient $\nabla \mathcal{L}_S(\boldsymbol{\eta}, \boldsymbol{\gamma})$ with the inverse of the Fisher information of the variational posterior $q$. See Appendix for the exact forms of the natural gradients with respect to $\boldsymbol{\eta}$ and $\boldsymbol{\gamma}$. To update the parameters $\boldsymbol{\eta}$ and $\boldsymbol{\gamma}$, we apply the stochastic natural gradient ascent rule

$$\boldsymbol{\eta}_{t+1} = \boldsymbol{\eta}_t + \rho_t \tilde{\nabla}_{\boldsymbol{\eta}} \mathcal{L}_S(\boldsymbol{\eta}_t, \boldsymbol{\gamma}_t), \quad \boldsymbol{\gamma}_{t+1} = \boldsymbol{\gamma}_t + \rho_t \tilde{\nabla}_{\boldsymbol{\gamma}} \mathcal{L}_S(\boldsymbol{\eta}_t, \boldsymbol{\gamma}_t), \tag{7}$$

where the step size is given by $\rho_t = \tau_0(\tau_1 + t)^{-\kappa}$. To ensure convergence, the $\tau_0, \tau_1, \kappa$ are set such that $\sum_t \rho_t^2 < \infty$ and $\sum_t \rho_t = \infty$ (Section 5 has our experimental values). The global update only costs $\mathrm{O}(NK)$ time per iteration due to the parsimonious $\mathrm{O}(K)$ parameterization of our PTM.

Our full inferential procedure is summarized in Algorithm 1. Within a mini-batch $S$, steps 2-3 can be trivially parallelized across triangles. Furthermore, the local parameters $q_{ijk}(x, y, z)$ can

be discarded between iterations, since all natural gradient sufficient statistics can be accumulated during the local update. This saves up to tens of gigabytes of memory on million-node networks.

## 5 Experiments

We demonstrate that our stochastic variational algorithm achieves latent space recovery accuracy comparable to or better than prior work, but in only a fraction of the time. In addition, we perform heldout link prediction and likelihood lower bound (i.e. perplexity) experiments on several large real networks, showing that our approach is orders of magnitude more scalable than previous work.

### 5.1 Generating Synthetic Data

We use two latent space models as the simulator for our experiments — the MMSB model [1] (which the MMSB batch variational algorithm solves for), and a model that produces power-law networks from a latent space (see Appendix for details). Briefly, the MMSB model produces networks with "blocks" of nodes characterized by *high edge probabilities*, whereas the Power-Law model produces "communities" centered around a *high-degree* hub node. We show that our algorithm rapidly and accurately recovers latent space roles based on these two notions of node-relatedness.

For both models, we synthesized ground truth role vectors $\theta_i$'s to generate networks of varying difficulty. We generated networks with $N \in \{500, 1000, 2000, 5000, 10000\}$ nodes, with the number of roles growing as $K = N/100$, to simulate the fact that large networks can have more roles [24]. We generated "easy" networks where each $\theta_i$ contains 1 to 2 nonzero roles, and "hard" networks with 1 to 4 roles per $\theta_i$. A full technical description of our networks can be found in the Appendix.

### 5.2 Latent Space Recovery on Synthetic Data

**Task and Evaluation.** Given one of the synthetic networks, the task is to recover estimates $\hat{\theta}_i$'s of the original latent space vectors $\theta_i$'s used to generate the network. Because we are comparing different algorithms (with varying model assumptions) on different networks (generated under their own assumptions), we standardize our evaluation by thresholding all outputs $\hat{\theta}_i$'s at $1/8 = 0.125$ (because there are no more than 4 roles per $\theta_i$), and use Normalized Mutual Information (NMI) [12, 23], a commonly-used measure of overlapping cluster accuracy, to compare the $\hat{\theta}_i$'s with the true $\theta_i$'s (thresholded similarly). In other words, we want to recover the set of non-zero roles.

**Competing Algorithms and Initialization.** We tested the following algorithms:

- **Our PTM stochastic variational algorithm.** We used $\delta = 50$ subsampling[2] (i.e. $\binom{50}{2} = 1225$ triangles per node), hyperparameters $\alpha = \lambda = 0.1$, and a 10% minibatch size with step-size $\tau_0(\tau_1 + t)^\kappa$, where $\tau_0 = 100$, $\tau_1 = 10000$, $\kappa = -0.5$, and $t$ is the iteration number. Our algorithm has a runtime complexity of $O(N\delta^2 K)$. Since our algorithm can be run in parallel, we conduct all experiments using 4 threads — compared to single-threaded execution, we observe this reduces runtime to about 40%.

- **MMTM collapsed blocked Gibbs sampler**, according to [8]. We also used $\delta = 50$ subsampling. The algorithm has $O(N\delta^2 K^3)$ time complexity, and is single-threaded.

- **PTM collapsed blocked Gibbs sampler**. Like the above MMTM Gibbs, but using our PTM model. Because of block sampling, complexity is still $O(N\delta^2 K^3)$. Single-threaded.

- **MMSB batch variational** [1]. This algorithm has $O(N^2 K^2)$ time complexity, and is single-threaded.

All these algorithms are locally-optimal search procedures, and thus sensitive to initial values. In particular, if nodes from two different roles are initialized to have the same role, the output is likely to merge all nodes in both roles into a single role. To ensure a meaningful comparison, we therefore provide the same fixed initialization to all algorithms — for every role $x$, we provide 2 example nodes $i$, and initialize the remaining nodes to have random roles. In other words, we seed 2% of the nodes with one of their true roles, and let the algorithms proceed from there[3].

**Recovery Accuracy.** Results of our method, `MMSB Variational`, `MMTM Gibbs` and `PTM Gibbs` are in Figure 1. Our method exhibits high accuracy (i.e. NMI close to 1) across almost all networks, validating its ability to recover latent roles under a range of network sizes $N$ and roles $K$. In contrast, as $N$ (and thus $K$) increases, `MMSB Variational` exhibits degraded performance despite having converged, while `MMTM/PTM Gibbs` converge to and become stuck in local minima

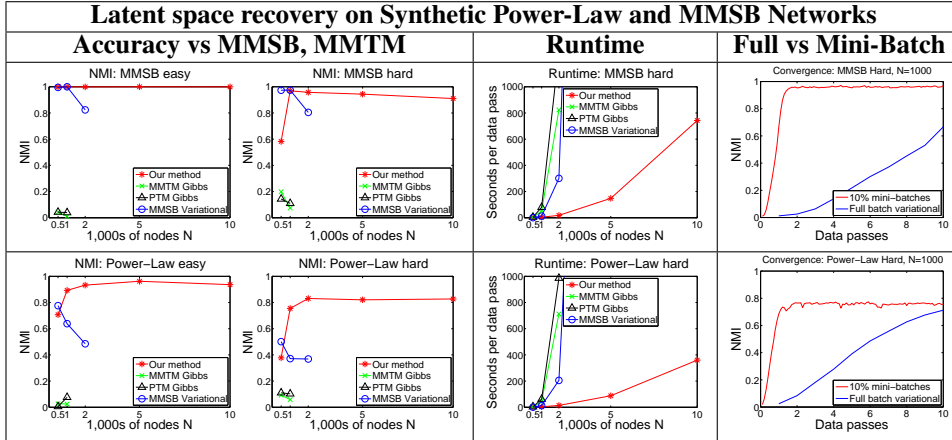

Figure 1: **Synthetic Experiments**. **Left/Center**: Latent space recovery accuracy (measured using Normalized Mutual Information) and runtime per data pass for our method and baselines. With the `MMTM/PTM Gibbs` and `MMSB Variational` algorithms, the larger networks did not complete within 12 hours. The runtime plots for *MMSB easy* and *Power-Law easy* experiments are very similar to the *hard* experiments, so we omit them. **Right**: Convergence of our stochastic variational algorithm (with 10% minibatches) versus a batch variational version of our algorithm. On $N = 1,000$ networks, our minibatch algorithm converges within 1-2 data passes.

| Link Prediction on Synthetic and Real Networks | | | | | | | | | |
|---|---|---|---|---|---|---|---|---|---|
| **Network Type** | | Synthetic | | Dictionary | | Biological | arXiv Collaboration | | Internet | Social |
| **Name** | MMSB | Power-law | Roget | Odlis | Yeast | GrQc | AstroPh | Stanford | Youtube |
| **Nodes** $N$ | 2.0K | 2.0K | 1.0K | 2.9K | 2.4K | 5.2K | 18.7K | 282K | 1.1M |
| **Edges** | 40K | 40K | 3.6K | 16K | 6.6K | 14K | 200K | 2.0M | 3.0M |
| **Our Method AUC** | **0.93** | **0.97** | 0.65 | 0.81 | 0.75 | **0.82** | **0.86** | **0.94** | **0.71** |
| **MMSB Variational AUC** | 0.91 | 0.94 | **0.72** | **0.88** | **0.81** | 0.77 | — | — | — |

Table 2: **Link Prediction Experiments,** measured using AUC. Our method performs similarly to `MMSB Variational` on synthetic data. MMSB performs better on smaller, non-social networks, while we perform better on larger, social networks (or MMSB fails to complete due to lack of scalability). Roget, Odlis and Yeast networks are from Pajek datasets (`http://vlado.fmf.uni-lj.si/pub/networks/data/`); the rest are from Stanford Large Network Dataset Collection (`http://snap.stanford.edu/data/`).

(even after many iterations and trials), without reaching a good solution[4]. We believe our method maintains high accuracy due to its parsimonious $O(K)$ parameter structure — compared to `MMSB Variational`'s $O(K^2)$ block matrix and `MMTM Gibbs`'s $O(K^3)$ tensor of triangle parameters. Having fewer parameters may lead to better parameter estimates, and better task performance.

**Runtime.** On the larger networks, `MMSB Variational` and `MMTM/PTM Gibbs` did not even finish execution due to their high runtime complexity. This can be seen in the runtime graphs, which plot the time taken per data pass[5]: at $N = 5,000$, all 3 baselines require orders of magnitude more time than our method does at $N = 10,000$. Recall that $K = O(N)$, and that our method has time complexity $O(N\delta^2 K)$, while `MMSB Variational` has $O(N^2 K^2)$, and `MMTM/PTM Gibbs` has $O(N\delta^2 K^3)$ — hence, our method runs in $O(N^2)$ on these synthetic networks, while the others run in $O(N^4)$. This highlights the need for network methods that are linear in $N$ and $K$.

**Convergence of stochastic vs. batch algorithms.** We also demonstrate that our stochastic variational algorithm with 10% mini-batches converges much faster to the correct solution than a non-stochastic, full-batch implementation. The convergence graphs in Figure 1 plot NMI as a function of data passes, and show that our method converges to the (almost) correct solution in 1-2 data passes. In contrast, the batch algorithm takes 10 or more data passes to converge.

### 5.3 Heldout Link Prediction on Real and Synthetic Networks

We compare `MMSB Variational` and our method on a link prediction task, in which 10% of the edges are randomly removed (set to zero) from the network, and, given this modified network, the task is to rank these heldout edges against an equal number of randomly chosen non-edges. For MMSB, we simply ranked according to the link probability under the MMSB model. For our

| Real Networks — Statistics, Experimental Settings and Runtime | | | | | | | | |
|---|---|---|---|---|---|---|---|---|
| Name | Nodes | Edges | $\delta$ | 2,3-Tris (for $\delta$) | Frac. 3-Tris | Roles $K$ | Threads | Runtime (10 data passes) |
| Brightkite | 58K | 214K | 50 | 3.5M | 0.11 | 64 | 4 | 34 min |
| Brightkite | \|\| | \|\| | \|\| | \|\| | \|\| | 300 | 4 | 2.6 h |
| Slashdot Feb 2009 | 82K | 504K | 50 | 9.0M | 0.030 | 100 | 4 | 2.4 h |
| Slashdot Feb 2009 | \|\| | \|\| | \|\| | \|\| | \|\| | 300 | 4 | 6.7 h |
| Stanford Web | 282K | 2.0M | 20 | 11.4M | 0.57 | 5 | 4 | 10 min |
| Stanford Web | \|\| | \|\| | 50 | 25.0M | 0.42 | 100 | 4 | 6.3 h |
| Berkeley-Stanford Web | 685K | 6.6M | 30 | 57.6M | 0.55 | 100 | 8 | 15.2 h |
| Youtube | 1.1M | 3.0M | 50 | 36.0M | 0.053 | 100 | 8 | 9.1 h |

Table 3: **Real Network Experiments**. All networks were taken from the Stanford Large Network Dataset Collection; directed networks were converted to undirected networks via symmetrization. Some networks were run with more than one choice of settings. *Runtime* is the time taken for 10 data passes (which was more than sufficient for convergence on all networks, see Figure 2).

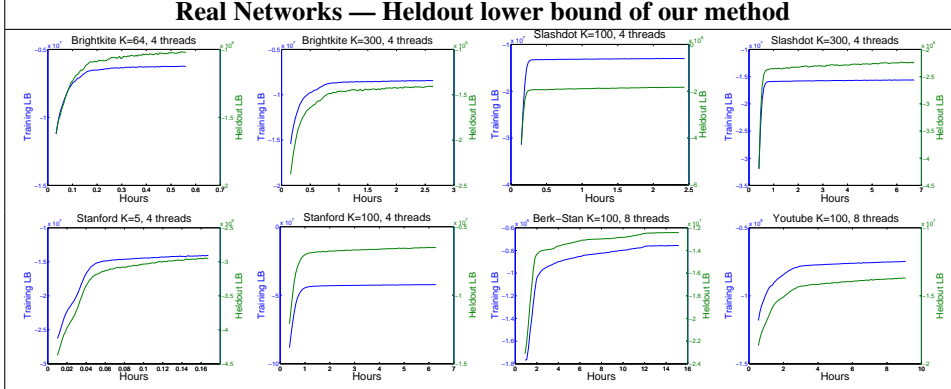

Figure 2: **Real Network Experiments**. Training and heldout variational lower bound (equivalent to perplexity) convergence plots for all experiments in Table 3. Each plot shows both lower bounds over 10 data passes (i.e. 100 iterations with $10\%$ minibatches). In all cases, we observe convergence between 2-5 data passes, and the shape of the heldout curve closely mirrors the training curve (i.e. no overfitting).

method, we ranked possible links $i - j$ by the probability that the triangle $(i, j, k)$ will include edge $i - j$, marginalizing over all choices of the third node $k$ and over all possible role choices for nodes $i, j, k$. Table 2 displays results for a variety of networks, and our triangle-based method does better on larger social networks than the edge-based MMSB. This matches what has been observed in the network literature [24], and further validates our triangle modeling assumptions.

### 5.4 Real World Networks — Convergence on Heldout Data

Finally, we demonstrate that our approach is capable of scaling to large real-world networks, achieving convergence in a fraction of the time reported by recent work on scalable network modeling. Table 3 lists the networks that we tested on, ranging in size from $N = 58$K to $N = 1.1$M. With a few exceptions, the experiments were conducted with $\delta = 50$ and 4 computational threads. In particular, for every network, we picked $\delta$ to be larger than the average degree, thus minimizing the amount of triangle data lost to subsampling. Figure 2 plots the training and heldout variational lower bound for several experiments, and shows that our algorithm always converges in 2-5 data passes.

We wish to highlight two experiments, namely the *Brightkite* network for $K = 64$, and the *Stanford* network for $K = 5$ (the first and fifth rows respectively in Table 3). Gopalan et al. ([5]) reported convergence on *Brightkite* in 8 days using their scalable a-MMSB algorithm with 4 threads, while Ho et al. ([8]) converged on *Stanford* in 18.5 hours using the MMTM Gibbs algorithm on 1 thread. In both settings, our algorithm is orders of magnitude faster — using 4 threads, it converged on *Brightkite* and *Stanford* in just 12 and 4 minutes respectively, as seen in Figure 2.

In summary, we have constructed a latent space network model with $O(NK)$ parameters and devised a stochastic variational algorithm for $O(NK)$ inference. Our implementation allows network analysis with millions of nodes $N$ and hundreds of roles $K$ in hours on a single multi-core machine, with competitive or improved accuracy for latent space recovery and link prediction. These results are orders of magnitude faster than recent work on scalable latent space network modeling [5, 8].

### Acknowledgments
This work was supported by AFOSR FA9550010247, NIH 1R01GM093156 and DARPA FA87501220324 to Eric P. Xing. Qirong Ho is supported by an A-STAR, Singapore fellowship. Junming Yin is supported by a Ray and Stephanie Lane Research Fellowship.

## Footnotes

[1] We tested a naive mean-field approximation, and it performed very poorly. This is because the tensor of role probabilities $q(x, y, z)$ is often of high rank, whereas naive mean-field is a rank-1 approximation.

[2] We chose $\delta = 50$ because almost all our synthetic networks have median degree $\leq 50$. Choosing $\delta$ above the median degree ensures that more than 50% of the nodes will receive all their assigned triangles.

[3] In general, one might not have any ground truth roles or labels to seed the algorithm with. For such cases, our algorithm can be initialized as follows: rank all nodes according to the number of 3-triangles they touch, and then seed the top $K$ nodes with different roles $x$. The intuition is that "good" roles may be defined as having a high ratio of 3-triangles to 2-triangles among participating nodes.

[4] With more generous initializations (20 out of 100 ground truth nodes per role), MMTM/PTM Gibbs converge correctly. In practice however, this is an unrealistic amount of prior knowledge to expect. We believe that more sophisticated MCMC schemes may fix this convergence issue with MMTM/PTM models.

[5]One data pass is defined as performing variational inference on $m$ triangles, where $m$ is equal to the total number of triangles. This takes the same amount of time for both the stochastic and batch algorithms.

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
