[Supplementary Material · Yin-Ho-Xing-NIPS13-supple.pdf]

# Appendix
# A Scalable Approach to Probabilistic Latent Space Inference of Large-Scale Networks

## A    Details of Stochastic Variational Inference

**Exact form of the variational lower bound.**    We adopted a structured mean-field approximation method, in which the true (but intractable) posterior of latent variables $p(\mathbf{s}, \boldsymbol{\theta}, \mathbf{B} \mid \mathbf{E}, \alpha, \lambda)$ is approximated by a *partially* factorized distribution $q(\mathbf{s}, \boldsymbol{\theta}, \mathbf{B})$,

$$q(\mathbf{s}, \boldsymbol{\theta}, \mathbf{B}) = q(\mathbf{s} \mid \boldsymbol{\phi})q(\boldsymbol{\theta} \mid \boldsymbol{\gamma})q(\mathbf{B} \mid \boldsymbol{\eta})$$

$$= \prod_{(i,j,k)\in I} q(s_{i,jk}, s_{j,ik}, s_{k,ij} \mid \phi_{ijk}) \prod_{i=1}^{N} q(\theta_i \mid \gamma_i) \prod_{x=1}^{K} q(B_{xxx} \mid \eta_{xxx}) \prod_{x=1}^{K} q(B_{xx} \mid \eta_{xx})q(B_0 \mid \eta_0), \quad (1)$$

where $I$ is the set of triples with triangular motifs formed: $I = \{(i,j,k) : i < j < k, E_{ijk} = 1, 2, 3 \text{ or } 4\}$. $|I| = O(N\delta^2)$ after $\delta$-subsampling.

The variational lower bound of the log marginal likelihood of the triangular motifs based on this variational distribution is

$$\log p(\mathbf{E} \mid \alpha, \lambda) \geq \mathbb{E}_q[\log p(\mathbf{E}, \mathbf{s}, \boldsymbol{\theta}, \mathbf{B} \mid \alpha, \lambda)] - \mathbb{E}_q[\log q(\mathbf{s}, \boldsymbol{\theta}, \mathbf{B})] \doteq \mathcal{L}(\boldsymbol{\phi}, \boldsymbol{\eta}, \boldsymbol{\gamma}) \quad (2)$$

$$= \mathbb{E}_q[\log p(B_0 \mid \lambda)] - E_q[\log q(B_0 \mid \eta_0)] + \sum_{x=1}^{K} \Big\{ \mathbb{E}_q[\log p(B_{xx} \mid \lambda)] - \mathbb{E}_q[\log q(B_{xx} \mid \eta_{xx})] \Big\}$$

$$+ \sum_{x=1}^{K} \Big\{ \mathbb{E}_q[\log p(B_{xxx} \mid \lambda)] - \mathbb{E}_q[\log q(B_{xxx} \mid \eta_{xxx})] \Big\} + \sum_{i=1}^{N} \Big\{ \mathbb{E}_q[\log p(\theta_i \mid \alpha)] - \mathbb{E}_q[\log q(\theta_i \mid \gamma_i)] \Big\}$$

$$+ \sum_{(i,j,k)\in I} \Big\{ \mathbb{E}_q[\log p(s_{i,jk} \mid \theta_i) + \log p(s_{j,ik} \mid \theta_j) + \log p(s_{k,ij} \mid \theta_k)] + \mathbb{E}_q[\log p(E_{ijk} \mid s_{i,jk}, s_{j,ik}, s_{k,ij}, \mathbf{B})] \Big\}$$

$$- \sum_{(i,j,k)\in I} \mathbb{E}_q[\log q(s_{i,jk}, s_{j,ik}, s_{k,ij} \mid \phi_{ijk})].$$

The first two line in (2) is the global term $g(\boldsymbol{\gamma}, \boldsymbol{\eta})$ that depends only the global variational paramters $\boldsymbol{\gamma}$ and $\boldsymbol{\eta}$, whereas the last two lines is a summation of local term $\ell(\phi_{ijk}, \boldsymbol{\gamma}, \boldsymbol{\eta})$, one for each triangular motif.

**Exact local update.** For each sampled triangle $(i,j,k)$ in a mini-batch, update the $O(K^3)$ entries of the tensor parameters $\phi_{ijk}$ as follows and then normalize to have sum equal to one.

- For $x \in \{1, \dots, K\}$,

$$\phi_{ijk}^{xxx} \propto \exp \Big\{ \mathbb{E}_q[\log B_{xxx,2}]\mathbb{I}[E_{ijk} = 4] + \mathbb{E}_q[\log(B_{xxx,1}/3)]\mathbb{I}[E_{ijk} \neq 4] + \mathbb{E}_q[\log \theta_{i,x}] + \mathbb{E}_q[\log \theta_{j,x}] + \mathbb{E}_q[\log \theta_{k,x}] \Big\}.$$
(3)

- For $x, y \in \{1, \dots, K\}$ and $x \neq y$,

$$\phi_{ijk}^{xxy} \propto \exp \Big\{ \mathbb{E}_q[\log B_{xx,3}]\mathbb{I}[E_{ijk} = 4] + \mathbb{E}_q[\log B_{xx,2}]\mathbb{I}[E_{ijk} = 3] + \mathbb{E}_q[\log(B_{xx,1}/2)]\mathbb{I}[E_{ijk} = 1 \text{ or } 2] \quad (4)$$

$$+ \mathbb{E}_q[\log \theta_{i,x}] + \mathbb{E}_q[\log \theta_{j,x}] + \mathbb{E}_q[\log \theta_{k,x}] \Big\}.$$

- For distinct $x, y, z \in \{1, \ldots, K\}$,

$$\phi_{ijk}^{xyz} \propto \exp \left\{ \mathbb{E}_q[\log B_{0,2}] \mathbb{I}[E_{ijk} = 4] + \mathbb{E}_q[\log(B_{0,1}/3)] \mathbb{I}[E_{ijk} \neq 4] + \mathbb{E}_q[\log \theta_{i,x}] + \mathbb{E}_q[\log \theta_{j,x}] + \mathbb{E}_q[\log \theta_{k,x}] \right\}. \quad (5)$$

The update equations for $\phi_{ijk}^{xyx}$ and $\phi_{ijk}^{yxx}$ are similar to $\phi_{ijk}^{xxy}$, and therefore we omit the details.

**Global update.** The natural gradient $\tilde{\nabla} \mathcal{L}_S(\boldsymbol{\eta}, \boldsymbol{\gamma})$ with respect to $\boldsymbol{\eta}$ is

- For $x \in \{1, \ldots, K\}$,

$$\tilde{\nabla}_{\eta_{xxx,1}} \mathcal{L}_S(\boldsymbol{\eta}, \boldsymbol{\gamma}) = \lambda + \frac{m}{s} \left[ \sum_{(i,j,k) \in S} q_{ijk}(x,x,x) \mathbb{I}[E_{ijk} \neq 4] \right] - \eta_{xxx,1}, \quad (6)$$

$$\tilde{\nabla}_{\eta_{xxx,2}} \mathcal{L}_S(\boldsymbol{\eta}, \boldsymbol{\gamma}) = \lambda + \frac{m}{s} \left[ \sum_{(i,j,k) \in S} q_{ijk}(x,x,x) \mathbb{I}[E_{ijk} = 4] \right] - \eta_{xxx,2}. \quad (7)$$

- For $x \in \{1, \ldots, K\}$,

$$\tilde{\nabla}_{\eta_{xx,1}} \mathcal{L}_S(\boldsymbol{\eta}, \boldsymbol{\gamma}) = \lambda + \frac{m}{s} \left[ \sum_{(i,j,k) \in S} \sum_{y:y \neq x} \Big( q_{ijk}(x,x,y) \mathbb{I}[E_{ijk} = 1,2] + q_{ijk}(x,y,x) \mathbb{I}[E_{ijk} = 1,3] \right. \quad (8)$$

$$\left. + q_{ijk}(y,x,x) \mathbb{I}[E_{ijk} = 2,3] \Big) \right] - \eta_{xx,1},$$

$$\tilde{\nabla}_{\eta_{xx,2}} \mathcal{L}_S(\boldsymbol{\eta}, \boldsymbol{\gamma}) = \lambda + \frac{m}{s} \left[ \sum_{(i,j,k) \in S} \sum_{y:y \neq x} \Big( q_{ijk}(x,x,y) \mathbb{I}[E_{ijk} = 3] + q_{ijk}(x,y,x) \mathbb{I}[E_{ijk} = 2] \right. \quad (9)$$

$$\left. + q_{ijk}(y,x,x) \mathbb{I}[E_{ijk} = 1] \Big) \right] - \eta_{xx,2},$$

$$\tilde{\nabla}_{\eta_{xx,3}} \mathcal{L}_S(\boldsymbol{\eta}, \boldsymbol{\gamma}) = \lambda + \frac{m}{s} \left[ \sum_{(i,j,k) \in S} \sum_{y:y \neq x} \Big( q_{ijk}(x,x,y) + q_{ijk}(x,y,x) + q_{ijk}(y,x,x) \Big) \mathbb{I}[E_{ijk} = 4] \right] - \eta_{xx,3}. \quad (10)$$

-

$$\tilde{\nabla}_{\eta_{0,1}} \mathcal{L}_S(\boldsymbol{\eta}, \boldsymbol{\gamma}) = \lambda + \frac{m}{s} \left[ \sum_{(i,j,k) \in S} \sum_{(x,y,z):x \neq y \neq z} q_{ijk}(x,y,z) \mathbb{I}[E_{ijk} \neq 4] \right] - \eta_{0,1}, \quad (11)$$

$$\tilde{\nabla}_{\eta_{0,2}} \mathcal{L}_S(\boldsymbol{\eta}, \boldsymbol{\gamma}) = \lambda + \frac{m}{s} \left[ \sum_{(i,j,k) \in S} \sum_{(x,y,z):x \neq y \neq z} q_{ijk}(x,y,z) \mathbb{I}[E_{ijk} = 4] \right] - \eta_{0,2}. \quad (12)$$

The natural gradient $\tilde{\nabla} \mathcal{L}_S(\boldsymbol{\eta}, \boldsymbol{\gamma})$ with respect to $\boldsymbol{\gamma}$ is, for each $i = 1, \ldots, N$ and $x = 1, \ldots, K$,

$$\tilde{\nabla}_{\gamma_{i,x}} \mathcal{L}_S(\boldsymbol{\eta}, \boldsymbol{\gamma}) = \alpha + \frac{m}{s} \left[ \sum_{(j,k):(i,j,k) \in S} \sum_{y,z} q_{ijk}(x,y,z) + \sum_{(j,k):(j,i,k) \in S} \sum_{y,z} q_{jik}(y,x,z) + \sum_{(j,k):(j,k,i) \in S} \sum_{y,z} q_{jki}(y,z,x) \right] - \gamma_{i,x}. \quad (13)$$

# B More Experimental Details

In the main paper, we omitted certain technical details about our experiments. For completeness, we shall furnish them here.

| Synthetic Data — Statistics for the largest ($N = 10,000$) networks | | | | | | |
|---|---|---|---|---|---|---|
| Network | Nodes $N$ | Edges $M$ | Degree mean/median/max | 2,3-Tris ($\delta = 50$) | Frac. of 3-Tris | Roles $K$ |
| MMSB easy | 10K | 279K | 55.9/56/81 | 11.0M | 0.060 | 100 |
| MMSB hard | 10K | 282K | 56.4/56/85 | 11.2M | 0.047 | 100 |
| Power-Law easy | 10K | 200K | 40/41/126 | 5.2M | 0.31 | 100 |
| Power-Law hard | 10K | 200K | 40/39/176 | 5.5M | 0.23 | 100 |

Table 1: **Synthetic Data Experiments**. Statistics for the largest ($N = 10,000$) networks.

## B.1 Generating Synthetic Data

**Latent Space Models.** We use two latent space models as the basis for our experiments — the MMSB model (Airoldi et al., 2009) (which the MMSB batch variational algorithm solves for), and a model that produces power-law networks from a latent space. A description of both models follows:

1. **MMSB:** Let $B$ be a $K \times K$ symmetric block matrix, the probability of an edge from $i$ to $j$ is $\theta_i^T B \theta_j$. We symmetrize the resulting network, converting all directed edges into undirected ones.

2. **Power-Law latent space model:** Let $M$ be the number of edges in the network. We generate all $M$ edges by repeating the following procedure: (a) pick a vertex $i$ with probability proportional to its degree; (b) draw a destination role $x \sim \text{Discrete}(\theta_i)$; (c) find the set $V_x$ of all vertices $v$ such that $\theta_{vx}$ is the largest element of $\theta_v$ (breaking ties at random); (d) within $V_x$, pick the destination vertex $j$ with probability proportional to its degree, and generate the undirected edge $(i, j)$. If $(i, j)$ is already present, we repeat the procedure.

The MMSB model produces networks with "blocks" of nodes characterized by *high edge probabilities*, whereas the Power-law model produces "communities" centered around a *high-degree* hub node. We show that our algorithm rapidly and accurately recovers latent space roles based on these two notions of node-relatedness.

**Ground Truth Role Vectors.** For both models, we synthesized ground truth role vectors $\theta_i$'s to generate networks of varying difficulty. We generated networks with $N \in \{500, 1000, 2000, 5000, 10000\}$ nodes, with the number of roles growing as $K = N/100$ (i.e. linear in $N$). We set the ground truth $\theta_i$'s as follows: first, we divided the nodes into $K$ groups of size 100. For the $x$-th group, we set 90 vectors $\theta_i$'s to have mass 1 in role $x$, i.e. $\theta_{ix} = 1$. The remaining 10 vectors $\theta_i$'s were set to have mass 0.5 in role $x$, and 0.5 in another randomly chosen role. This forms a latent space where $90\%$ of the nodes have pure-membership, and $10\%$ have mixed-membership between 2 roles. We call these networks "MMSB easy" and "Power-Law easy", respectively.

We also created a second, more challenging series of networks (we call them "hard") using role vectors with heavier mixing. These roles were constructed as follows: for the $x$-th group, we set 80 vectors $\theta_i$'s to have mass 1 in role $x$, 10 vectors $\theta_i$'s to have 0.5 mass in role $x$ and 0.5 mass in 1 other random role, and 10 vectors $\theta_i$'s to have 0.25 mass in role $x$ and 0.25 mass in 3 other random roles. The resulting latent space has nodes with up to 4 roles.

In total, we generated 20 networks: 5 sizes $\times$ 2 models $\times$ 2 sets of role vectors; summary statistics for the 4 largest $N = 10,000$ networks can be found in Table 1. For networks under the Power-Law model, we generated $M = 20N$ edges (so the average degree is 40). As for networks under the MMSB model, we used a block matrix $B$ with diagonal elements set to 0.2, and off-diagonal elements set to 0.001. Under this $B$, the ratio of intra-role to inter-role edges decreases as $(N, K)$ increase — from approximately $20:1$ at $(N = 1000, K = 10)$, to $2:1$ at $(N = 10000, K = 100)$. In this sense, the amount of noise increases as the network gets larger, making membership recovery harder.