[Reviews · NeurIPS 2013]

Submitted by Assigned_Reviewer_3

This paper proposes parsimonious triangular model (PTM), which constrains the O(K^3) parameter space of mixed-membership triangular model (MMTM) to O(K) for faster inference. Authors develop a stochastic variational inference algorithm for PTM and additional approximation tricks to make it further scalable. It is shown from synthetic dataset that the reduction of the number of variables may lead to stronger statistical power, and from real-world datasets that the proposed method is competitive with existing methods in terms of accuracy.

Quality:

PTM seems to be an interesting specialization of MMTM, but it is questionable what is the practical advantage of achieving good scalability in terms of K (the number of possible roles). To empirically evaluate the value of such a method, it is critical for us to answer "how does it help if we can learn MMTM with large K?" Since MMSB and MMTM are mixed-membership models, using small K may not be as troublesome as it is in single-membership models! For real network experiments, what would be really interesting is to see the performance of both MMTM and PTM as a function of K: if we can really see that PTM with large K outperforms MMTM with small K, it will show the power of PTM. Unfortunately such experiments were not done. Actually I could not find good description about how K's were chosen in real data experiments for both PTM and its competitor MMSB; the method of model selection could've biased experimental results.

Authors are also a bit reticent on what are the impact of simplifying assumptions they make. It is clear that we can achieve good scalability by constraining the parameter space, but it would be also nice to hear about when would such model fail. If there really are O(K^3) distinct values in the parameter matrix B (following authors' notation) of MMSB and MMTM, PTM should perform poorly, right? In this regard, it is unsatisfactory that the choice of true B in synthetic data experiments were not clearly described; authors' should've experimented with various B's which some are in favor of PTM and some are against the model. It would be more beneficial for readers to know when would PTM succeed and when would it fail, than to know only when it would succeed.

On the other hand, authors introduce a number of tricks to make PTM scale; while I find them to be very interesting and effective, this paper does a poor job in describing and analyzing the impact of such an approximation. 1) What is the impact of choosing the threshold \delta? On real-world and synthetic data experiments, does it help to choose larger or smaller \delta? How much can we lower \delta to improve scalability while not sacrificing much of accuracy? 2) Authors claim that a-MMSB does not work since downsampling compromises accuracy; why would PTM free of such problem? MMTM and PTM are actually ignoring triplets with only one edge. Why can't we ignore non-edges in MMSB while we can in PTM? 3) What is the impact of O(K) approximation to local update (in page 5)? What happens if we sample 2K, 4K, or 100K of triples? Would the quality of approximation suffer when K gets larger?

To sum up, I am not convinced experimental setup was fair to competitors, and the description of potential weakness of the algorithm/model is poor.

Clarity: the paper is very clearly written and well-organized.

Originality: In terms of modeling it is a bit incremental, but achieving scalability via constraining parameters is an interesting and original direction to pursue, as recently in this topic a lot of work has been focused on generalizing existing models.

Significance: Provided simplifying assumptions in the model and the inference algorithm do hold in practice, it would have a good impact on the practice of latent-space models on networks, as not many of them actually scale to large data.

Additional Comments: Personally I think the title is too broad. There are other scalable approaches to other latent space models, and authors propose a scalable algorithm to _a_ latent space model- PTM.

Also, I am not sure that PTM is really a probabilistic method. Authors argue in the last paragraph of Section 3 that this is same in spirit to bag-of-words assumption, but I am not convinced. When LDA generates a bag-of-words, there always exists a document corresponding to it; so it has proper likelihood function, no matter how unrealistically simple it might be. On the other hand, with very high probability there does not exist any network corresponding to a bag-of-triples generated by MMTM and PTM. Therefore MMTM/PTM is not really a generative model for networks, and majority of posterior probability mass is placed on modeling non-networks! It might be dangerous to abandon basic principles of likelihood in "probabilistic" methods (what does probability mean in PTM?)... I am curious what authors and other reviewers think.
Summary: Authors propose modeling assumptions and inference tricks that would make inference on MMTM scalable. Side-effects of such decision, however, are poorly investigated.

Submitted by Assigned_Reviewer_4

This paper proposes a technique for learning latent structure in social network data. The model is fit to triangle count statistics obtained from a preprocessing step, analogous to a bag-of-words representation for text documents. The main modeling advance is motivated by aMMSB: rather than parameterize all between-"community" interactions they consider the configurations of the triangles to identify a set of K roles to parameterize. This results in fewer parameters in the model and, in turn, a more scalable implementation. They apply recent advances stochastic variational inference -- deriving a global update and local updates for the proposed model.

The novel modeling contribution is a straightforward combination of the aMMSB with the MMTM, though somewhat tedious bookkeeping was required. The presentation of this material could be improved.

Though most of the learning algorithm is a direct application of stochastic variational inference, the authors provide an intuitive and intriguing approximation for the local update rule.

Given the prevalence of directed networks, a detailed discussion of how these methods might extend would be welcome.

Ongoing concerns:
- "our Parsimonious Triangular Model (PTM) is essentially a latent space counterpart to ERGMs, that represents the processes of generating triangular motifs". This seems like a curious statement: 1) ERGMs can include much more varied statistics than just triangles and 2) I don't see how the PTM is a "latent space counterpart", unless all latent space network models are "latent space counterparts".
- Are the inferred latent space interpretable? If so, what do we learn about the similarities and differences among these graphs?
The difference between Delta^1 and Delta^2 is not clear (at least to this reader).
- It is not made clear (in this paper) why subsampling the triangles for each node (using delta) provides an adequate approximation.
- If the goal is to recover the "set of non-zero roles" then perhaps there should be more of a binary notion of "role" in the model.
- How robust are the latent space recovery results to the threshold of .125 (which seems somewhat arbitrary)?
- A variety of latent space models can produce power-law networks. Which one did you use for the experiments?
- Why is PTM CGS have K^3 complexity? Didn't you argue that there are O(K) parameters rather than O(K^3)?
- One concern is how this will perform when fewer edges are observed. In the experiments they only hold out 10% of the edges, presumably because the distribution of adjacent triangles will be quite biased if too many edges are removed prior to the preprocessing step.
- It is counterintuitive that the method beats MMSB on a synthetic network generated via MMSB (Figure 1, left 4 panels).
Summary: This work uses stochastic variational inference to learn a mixed-membership blockmodel (with a restricted set of parameters) on the triangle counts in a graph. Their modeling choices and approximations allow for a more scalable O(NK) algorithm.

Submitted by Assigned_Reviewer_5

The authors develop a scalable approximate inference algorithm for the
"triangle model" of networks. They develop a powerful subclass of the
model that shares parameters in a clever way, and they use stochastic
variational inference to infer the posterior with very large networks.
The paper is well written, and well exexcuted. I think it is acceptable.

p1: What is "parsimonious" modeling? I don't think this is a
run-of-the-mill expression in the context of probabilistic models. (I
might be wrong about that.) I suggest unpacking that term in a
sentence since it is important to the paper.

p1: From my reading of Gopalan et al., down-sampling the zeros does
not comprimise accuracy. Your sentence in the introduction is
misleading.

p4: I'm sympathetic to the triangle model being OK from a pragmatic
perspective, but I don't think appealing to LDA is fair. If we run the
generative process of LDA, we will obtain something that "is" a
document, albeit a silly one, even if it's not the original. Running
your model, however, can produce objects that are not networks. Again,
this does not bother me---it's a good and practical model---but the
defense by means of LDA is not appropriate.

p5: You should cite Hoffman et al. with Amari and Sato. In your
scenario, where there are "local variables" (in Hoffman's language),
Sato's algorithm does not apply. The needed theory is in Hoffman.

p6: How did you generate power law networks in the latent space? The
paper is vague around this point.

p6: Earlier you cite Gopalan et al. as a way of speeding up MMSB. You
should compare to their algorithm, in addition to (or instead of) MMSB
batch.

Small comments

p2: mixture-membership -> mixed-membership

p8: I suggest removing the last sentence. The second-to-last sentence
is a stronger way to end the paper.

In the bibliography, cite the JMLR version of "Stochastic Variational
Inference".
Summary: The authors develop a scalable approximate inference algorithm for the
"triangle model" of networks. They develop a powerful subclass of the
model that shares parameters in a clever way, and they use stochastic
variational inference to infer the posterior with very large networks.
The paper is well written, and well exexcuted. I think it is acceptable.
Author Feedback

Author rebuttal: We thank all reviewers for their valuable comments and appreciation of our work's clarity, originality and significance. We also gladly acknowledge R4 & R5 for sharing our view on how the paper should be received. For networks with large scale N>1m and many overlapping communities K>100 (Yang & Leskovec, 2012; snap.stanford.edu), having O(NK) runtime is crucial. We have developed a highly scalable O(NK) technique targeted at such networks, enabling latent space inference at unprecedented scale (N>1m, K>100), while the state-of-the-art (MMTM, a-MMSB) only scales to N~=100k. Our algorithm delivers a much-needed technique for meaningful inference on large networks, while small N, K are not the target of our work.

R3:

There exist networks with many overlapping/mixed communities K (Yang & Leskovec, 2012), for which algorithms must scale in K. MMSB variational & MMTM CGS have O(K^2) & O(K^3) runtime, too slow for large K (Fig 1). Since PTM is intended for the large K setting, comparing large-K PTM against small-K MMTM is beyond the scope of this paper. K can be tuned via cross-validation on the heldout likelihood, though it is not our focus. In our link-prediction experiments on real world networks, the same K=20 (the MMSB computational limit from Fig 1) was used for MMSB and PTM, so there is no model selection bias.

The reviewer suggested the experiments favor PTM. This is untrue. The true B's for the synthetics are in the appendix (p3): “... under the MMSB model, we used B with diagonal elements 0.2, and off-diagonal elements 0.001”. This matches the MMSB generative assumption while mismatching PTM, so it actually favors MMSB. The other setting (power-law) does not favor any methods. Both synthetic and real experiments show MMSB variational beats PTM on small networks (Fig. 1, Table 2); we have shown PTM’s weaknesses.

One cannot experiment with O(K^3) distinct values in PTM’s B tensor, because PTM is a probabilistic model for triangular motifs, not a network simulation tool. Per conventional wisdom, we believe that relying on a simple, parsimonious model leads to better speed and generalization on large data.

Approximation quality is important, but is not the focus of this paper (some issues were addressed in Ho et. al., 2012). Our paper is about scalable network modeling rather than general-purpose approximate inference; such preferential focus exists in many variational approximation papers, e.g. mean field for Ising models and LDA. Hence we devoted the limited space to explaining our scalable approximation scheme (rather than its tuning details), while providing substantial evidence that it works well on synthetic and real networks. Moreover, we provide some guidance on choosing delta (footnote 2, p6), while the trade-off between scalability and accuracy was investigated by Ho et. al. (2012).

We did not mean to claim “a-MMSB does not work”; we meant there might be a variance issue with the down-sampling strategy, that can compromise accuracy (see R5 response). We also wish to clarify that down-sampling and ignoring 0-edges are not the same: the former means giving preferential attention to the subset of node pairs with edges when forming node pair mini-batches in the SVI algorithm; the latter means 0-edges are simply not modeled. a-MMSB can downsample 0-edges, but does not ignore them. PTM models the distribution of open and closed triangles, while ignoring node triples with one or no edges. The key insight is that ignoring node triples with 0/1 edges leaves two types of triples (2/3 edges), whose utility was justified in Ho et. al. (2012). However, ignoring 0-edges in MMSB leaves us with only 1-edges, leading to the B matrix being estimated as all 1's, an uninteresting result.

R4:

The ERGM statement was admittedly obscure. We meant that PTM is inspired by ERGM triangle count statistics. We will fix these comments.

The PTM latent space is interpretable: as a starting point, compare ratios of closed to open triangles in each B_xxx. A high ratio implies a legitimate, well-connected community, while a low ratio may signify a bipartite subgraph (e.g. scammer network).

The delta on p5 is the variational parameter; it is unrelated to the subsampling delta (p2). delta^2 (p6) is the subsampling delta squared.

delta-subsampling approximately preserves local cluster properties at each node, allowing much faster inference at a small accuracy cost (Ho et. al., 2012).

The NMI metric requires binary roles. The 0.125 threshold is half of 1/4 (because our synthetics use <=4 roles per node). We also used mixed roles, in held-out link prediction.

Power-law model: see appendix and R5 response.

Like MMTM CGS, PTM CGS block-samples variables s (not B, which is size O(K)), hence O(K^3) runtime.

Holding out 10% edges for prediction is typical (p7, Gopalan et. al., 2012). Each node gets approx. (1-10%)^2=81% of adjacent triangles.

On large networks, MMSB has far more latent variables than PTM, making inference harder and yielding poor results. On small networks, MMSB performs better (Fig. 1, Table 2).

R5:

The down-sampling statement was admittedly misleading. We meant that though the stochastic gradient expectation remains equal to the true gradient, the variance might become worse and compromise accuracy. We will fix the statement.

We did not mean to defend PTM by appealing to LDA; instead, we wanted to motivate PTM by drawing an analogy between our bag-of-triangles network representation and the bag-of-words document representation. We will fix those statements.

Thanks for pointing out Hoffman et al.’s theory, and we will gladly cite its JMLR version.

Details for the power-law model are in the appendix (p3). It closely follows the MMTM paper (Ho et. al., 2012).

Because a-MMSB has no public code, while the paper had insufficient details (e.g. initialization strategy) to fairly reproduce the algorithm, we could not perform a fair comparison.